# Prevalence and serotypes of Shiga toxin-producing *Escherichia coli* (STEC) in dairy cattle from Northern Portugal

Andressa Ballem[1,2,3,4,5], Soraia Gonçalves[1], Isidro Garcia-Meniño[6,7], Saskia C. Flament-Simon[6,7], Jesús E. Blanco[6,7], Conceição Fernandes[5], Maria José Saavedra[2,4], Carlos Pinto[1¤], Hugo Oliveira[8]*, Jorge Blanco[6,7], Gonçalo Almeida[1], Carina Almeida[1]*

**1** National Institute for Agrarian and Veterinary Research, Vairão, Vila do Conde, Portugal, **2** Veterinary Science Department, University of Trás-os-Montes and Alto Douro, Vila Real, Portugal, **3** Farroupilha Federal Institute, Campus of São Vicente do Sul, Rio Grande do Sul, São Vicente do Sul, Brazil, **4** Centre for the Research and Technology of Agro-Environmental and Biological Science, University of Trás-os-Montes and Alto Douro, Vila Real, Portugal, **5** Centro de Investigação de Montanha, School of Agriculture, Polytechnic Institute of Bragança, Bragança, Portugal, **6** Laboratorio de Referencia de *E. coli*, Department of Microbiology and Parasitology, Veterinary Faculty, University of Santiago de Compostela, Lugo, Spain, **7** Instituto de Investigación Sanitaria de Santiago de Compostela, Santiago de Compostela, Spain, **8** Centre of Biological Engineering, University of Minho, Campus of Gualtar, Braga, Portugal

¤ Current address: Faculdade de Ciências Agrárias e do Ambiente, University of Azores, Angra do Heroísmo, Portugal
* carina.almeida@iniav.pt (CA); hugooliveira@deb.uminho.pt (HO)

**Data Availability Statement:** All relevant data are within the manuscript and its Supporting Information files.

## Abstract

The prevalence of Shiga toxin (Stx)-producing *Escherichia coli* (STEC) was determined by evaluating its presence in faecal samples from 155 heifers, and 254 dairy cows in 21 farms at North of Portugal sampled between December 2017 and June 2019. The prevalence of STEC in heifers (45%) was significantly higher than in lactating cows (16%) ($p<0.05$, Fisher exact test statistic value is <0.00001). A total of 133 STEC were isolated, 24 (13.8%) carried Shiga-toxin 1 (*stx1*) genes, 69 (39.7%) carried Shiga-toxin 2 (*stx2*) genes, and 40 (23%) carried both *stx1* and *stx2*. Intimin (*eae*) virulence gene was detected in 29 (21.8%) of the isolates. STEC isolates belonged to 72 different O:H serotypes, comprising 40 O serogroups and 23 H types. The most frequent serotypes were O29:H12 (15%) and O113:H21 (5.2%), found in a large number of farms. Two isolates belonged to the highly virulent serotypes associated with human disease O157:H7 and O26:H11. Many other bovine STEC serotypes founded in this work belonged to serotypes previously described as pathogenic to humans. Thus, this study highlights the need for control strategies that can reduce STEC prevalence at the farm level and, thus, prevent food and environmental contamination.

## Introduction

Shiga toxin-producing *Escherichia coli* (STEC) are a heterogeneous group of foodborne pathogens with various levels of virulence for humans and are defined by the presence of one or both phage-encoded Shiga toxin genes: Shiga-toxin 1 (*stx1*) and Shiga-toxin 2 (*stx2*) [1, 2].

**Funding:** This study was financially supported by: i) project PhageSTEC (POCI-01-0145 -FEDER-029628) funded by FEDER through COMPETE2020 (Programa Operacional Competitividade e Internacionalização) and by National Funds thought FCT (Fundação para a Ciência e a Tecnologia); ii) strategic project UIDB/04469/2020 unit and BioTecNorte operation (NORTE-01-0145-FEDER-000004) funded by FCT under the scope of the European Regional Development Fund (Norte2020 - Programa Operacional Regional do Norte); iii) project PI16/01477 from Plan Estatal de I+D+I 2013-2016, Instituto de Salud Carlos III (ISCIII), Subdirección General de Evaluación y Fomento de la Investigación, Ministerio de Economía y Competitividad (Gobierno de España) and FEDER; and iv) grant ED431C2017/57 from the Consellería de Cultura, Educación e Ordenación Universitaria, (Xunta de Galicia) and FEDER; UIDB/AGR/04033/2020 by National Funds thought FCT. Author IGM acknowledges the Consellería de Cultura, Educación e Ordenación Universitaria, Xunta de Galicia for the individual grant ED481A-2015/149 and and author SCFS acknowledges the FPU programme for the individual grant FPU15/02644 from the Secretaría General de Universidades, Spanish Ministerio de Educación, Cultura y Deporte. The funders had no role in study design, data collection and analysis, decision to publish, or preparation of the manuscript.

**Competing interests:** The authors have declared that no competing interests exist.

STEC cause clinical illness in humans, ranging from uncomplicated non-bloody diarrhoea to severe diseases, such as acute gastroenteritis, haemorrhagic colitis (HC) and the life-threatening haemolytic uremic syndrome (HUS) [3, 4].

In 2018, 8161 human cases of STEC infections were confirmed in Europe. In fact, in the European Union, STEC ranks the third place on the most relevant foodborne pathogens, behind *Campylobacter* spp. and *Salmonella* spp. [5].

The *stx* genes are responsible for producing the Shiga toxins (Stx), also called Verotoxins. These toxins are grouped into two types, Stx1 and Stx2, each one including several variants that contribute for different virulence degrees. Subtypes within Stx1 include Stx1a, Stx1c and Stx1d, while for Stx2 seven subtypes (Stx2a, Stx2b, Stx2c, Stx2d, Stx2e, Stx2f, Stx2g) are well-established among the scientific community [6, 7]. Additionally, two new variants of Stx2 (Stx2h and Stx2k) have been recently identified [8, 9].

There are other virulence factors expressed by STEC such as, the protein intimin, coded by *eae* gene, that is associated with the capacitive of STEC to colonize the human gut and cause illness [10–12]. The intimin is responsible for intimate attachment of STEC to intestinal epithelial cells, causing attaching and effacing lesions in the intestinal mucosa. It is located at the pathogenicity island known as the locus of enterocyte effacement (LEE) [13].

The main STEC serotype associated with outbreaks and serious diseases in humans is O157:H7. However, the increase of outbreaks caused by non-O157 has given a warning signal to the virulence potential of other serotypes [2, 14].

STEC are found in a wide variety of animal species, as a natural inhabitant of gut. Ruminants, including cattle, are the most important reservoir of the zoonotic STEC. These pathogens can be transmitted to humans through many different routes, but contaminated food, by contact with faecal material, has been described as the main transmission route [15–18].

Beef and milk are the main animal food products responsible for outbreaks reported between 1998 and 2016 [19]. Because of this, the surveillance on the prevalent STEC serotypes, and *stx* subtypes, in cattle gut, is crucial to prevent transmission to humans and to design tailored control strategies against STEC.

In Portugal, there are no reports on the prevalence of STEC in dairy cattle and other ruminants. So, in this study we investigate the prevalence of STEC in healthy dairy cattle (lactating cows and heifers). Isolates were screened for the presence of major virulence genes and serotypes were determined.

## Materials and methods ethics statement

The study has been validated by a specialized panel composed by a veterinary physician, and two authorized persons to perform animal experimentation, with accreditation number 020/08 by FELASA—Federation of Laboratory Animal Science Associations. Animals were enrolled under the permission of farms' owners. No additional approvals were required by the authority as samples were collected during a standard procedure of rectal examination performed by a veterinary physician, which is an integral part of a thorough clinical examination in cows. This study was conducted in accordance with the E.U. Animal Welfare Directives (Directive 98/58/CE and Decreto-lei no 64/2000).

## Farms selection and collection of samples

Milk farms enrolled in this study were from the north-west region. This particular area, placed at "Entre Douro e Minho", has a high frequency of dairy herds and is usually referred to as the "Bacia leiteira primária". Farms were distributed in 12 different counties: Barcelos, Vila do Conde, Póvoa de Varzim, Ponte de Lima, Guimarães, Vila Nova de Cerveira, Chaves, Miranda

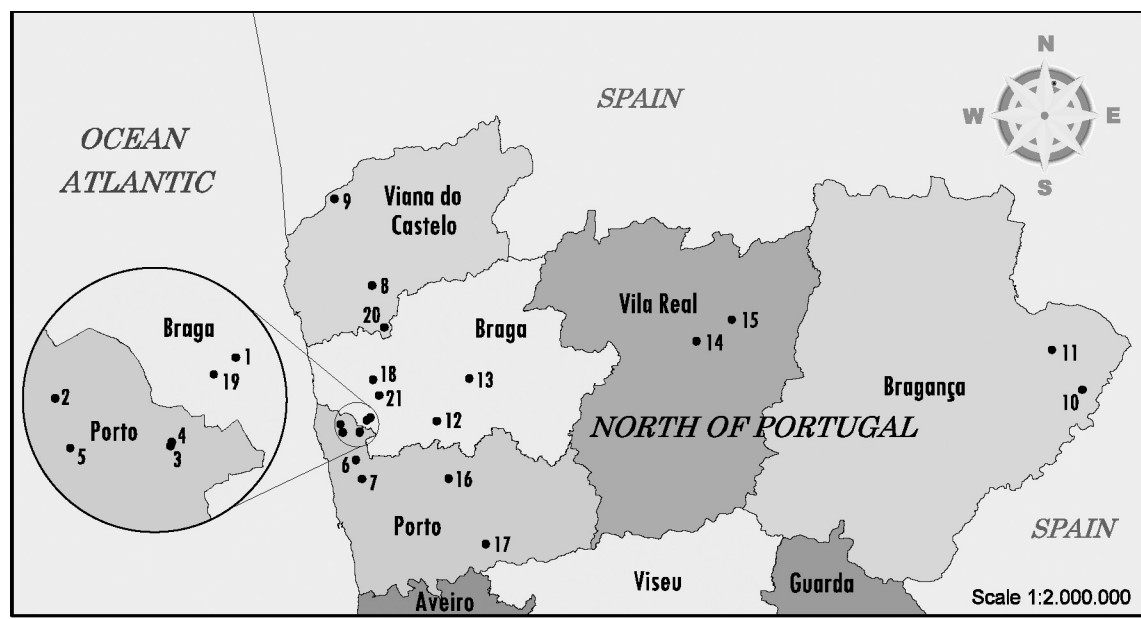

**Fig 1. Map of North of Portugal showing the dairy farms sampled for STEC screening.** The farm location and number of sampled animals can be consulted in S1 Table. Map was constructed using open data (public domain) from the Portuguese government that have Creative Commons Attribution 4.0 (CC BY 4.0) and was available at https://dados.gov.pt/pt/datasets/concelhos-de-portugal/ access on 12 February.

do Douro, Vimioso, Povoa de Lanhoso, Penafiel and Paços de Ferreira (Fig 1). Only farms with a documented herd size of more than 50 dairy cows or heifers were sampled. A total of 21 milk farms (S1 Table) were enrolled and faecal samples were collected randomly directly from the rectum of healthy animals, between December 2017 and June 2019. Farms distribution is presented in Fig 1. Map was created using MicroStation v8 (Bentley Systems, Incorporated). All farms kept animals indoors, in open stables, but, at times, animals were given the opportunity to graze outside. The animals were categorized into two groups: lactating dairy cows (more than 24 months of age) and heifers (age between 6 and 18 months). All animals sampled were of the Holstein-Friesian breed.

The minimum sample size of about 300 animals was calculated using the standard formula described by Thrusfield et al. [20] with confidence level of 0.95 and a margin of error of 0.05. Also, it was based on the assumption of a 27% prevalence of carriage described by Blanco *et al.* (2004) [12] in a prevalence study, in the neighbouring country Spain, in cattle samples collected between 1993 and 1995. A sample of 10% of the animals (lactating cows and heifers) was collected whenever possible at each farm, ranging from a minimum of 5 animals to a maximum of 20 per farm. A total of 409 samples from 254 (62.1%) adults lactating dairy cow, and 155 (37.9%) heifers, were collected and transported to the laboratory in portable, insulated cold boxes. The samples were kept at 4°C and examined within 24h.

## Isolation/ detection of STEC

The method used for isolation/ detection of STEC was adapted from ISO/TS 13136:2012 (E) [21]. Ten grams of faeces were placed in 90 mL of modified Tryptone Soy Broth media supplemented with 20 mg. $L^{-1}$ of novobiocin (mTSB+N), homogenized in a blend bag with filter and incubated at 37°C for 18 h to 24 h. Enrichment broth was subjected to multiplex real-time PCR to detect *stx* genes. Enrichment broths positive for *stx* genes were streaked onto MacConkey Sorbitol agar (SMAC) and Tryptone bile x-glucuronide medium (TBX) and incubated at

37˚C for 18 h to 24 h. Then, 50 single colonies with *E. coli* morphology were picked up from TBX and SMAC and point-inoculated on nutrient agar (NA). Plates were incubated at 37˚C for 18 h to 24 h. The 50 colonies were organized in 5 pools of 10 colonies each, and pools were then subjected to real-time PCR as described below. For pools positive to *stx* genes, the single colonies were analysed individually again. Isolates containing one or more *stx* genes were preserved at -80˚C. For pools with several *stx*-positive colonies, but with the same profile, only one colony was selected for storage and further analysis.

## DNA template preparation

DNA extraction from enrichment was performed according to the U. S. Food, Drug & Administration in Bacteriological Analytical Manual: Diarrheagenic *Escherichia coli* [22]. Briefly, 1 mL of overnight enrichment in mTSB+N was transferred to a microtube and centrifuged (12,000 × g) for 3 min. The pellet was washed with 1 ml 0.85% NaCl and centrifuged again at 12,000 × g for 3 min. The pellet was suspended in 1 ml of sterile ultrapure water. The microtubes were placed in a heat block at 100˚C for 15 min and then centrifuged 12,000 × g for 1 minute. The supernatant was diluted 10-fold in ultrapure sterile water.

Pure cultures (including control cultures) and pools from agar plate were suspended in microtubes with 1 mL of sterile ultrapure water, placed in a heat block at 100˚C for 15 min and centrifuged 12,000 × g for 1 minute. The supernatant was then used for real-time PCR screening. This procedure for template preparation has been previously tested in enriched faeces samples, artificially inoculated with STEC, to assure an appropriate performance in the following PCR-screening steps.

## Detection of *stx1*, *stx2* and *eae* in faecal samples by multiplex real time-PCR

DNA samples from enrichments, pools and single colonies were screened by multiplex real-time PCR for detection of *stx1*, *stx2* and *eae* gene sequences. This was performed according to ISO/TS 13136:2012(E) [21]. Primers, probes and the predicted lengths of PCR amplification products are listed in Table 1. Plasmid pUC19 was used as internal amplification control to

**Table 1. Primers and probes used for 5'-nuclease real-time-PCR assays[a].**

| Primer | Sequence 5'> 3'[b] | Amplicon size (bp) |
|---|---|---|
| F-*stx1* | TTTGTYACTGTSACAGCWGAAGCYTTACG | 131 |
| R-*stx1* | CCCCAGTTCARWGTRAGRTCMACRTC | |
| *stx1* probe | [FAM]CTGGATGATCTCAGTGGGCGTTCTTATGTAA[BHQ1] | |
| F-*stx2*[c] | TTTGTYACTGTSACAGCWGAAGCYTTACG | 128 |
| R-*stx2*[c] | CCCCAGTTCARWGTRAGRTCMACRTC | |
| *stx2* probe | [HEX]TCGTCAGGCACTGTCTGAAACTGCTCC[BHQ2] | |
| F-*eae* | CATTGATCAGGATTTTTCTGGTGATA | 102 |
| R-*eae* | CTCATGCGGAAATAGCCGTTA | |
| *eae* probe | [CY5]ATAGTCTCGCCAGTATTCGCCACCAATACC[BHQ2] | |
| **F-pUC19** | GCAGCCACTGGTAACAGGAT | 119 |
| **R-pUC19** | GCAGAGCGCAGATACCAAAT | |
| **pUC19 probe** | [ROX]AGAGCGAGGTATGTAGGCGG[BHQ2] | |

[a]Table adapted from ISO/TS 13136:2012 (E).

[b]In the sequence Y is (C, T), S is (C, G), W is (A, T), R is (A, G), M is (A, C).

[c]This combination of primer/probe recognizes all the *stx*2 variants except the *stx*2f.

monitor any possible inhibitory effect. PCR assays were carried out in a 25 μL volume containing: 2.5 μL of nucleic acid template, 12.5 μL of commercial real-time PCR Probe Master Mix (2x) (NZYtech), 1 μL of pUC 19 DNA (approximately 100 copies), 0.4 pmol. μL$^{-1}$ of each primer and 0.2 pmol. μL$^{-1}$ of each probe. DNA samples from *E. coli* O26 (LMV_E_2, INIAV culture collection), was used as a positive control. Negative controls were included in each run, where template was replaced by sterile ultrapure water. Temperature conditions consisted of an initial 95°C denaturation step for 5 min followed by 39 cycles at 95°C for 10 s (denaturation) and 60°C for 50 s (annealing and extension). For samples with evidence of inhibition (no amplification of the internal control, pUC19), DNA samples were diluted 1/10 and retested.

## Screening for virulence genes by conventional PCR

Conventional PCR was used for confirming the presence of *stx1*, *stx2* and *eae* genes. These screening was performed according to Mora *et al.* (2011) [23]. Primers, PCR conditions and the predicted lengths of PCR amplification products are listed in S2 Table. Multiplex PCR assays for *stx1* and *stx2* were carried out in a 25 μL volume containing, 5 μL of nucleic acid template, 12.5 μL of commercial Multiplex PCR Master Mix 2x (NZYtech), 1 μL (0.8 pmol. μL$^{-1}$) each primer and 2.5 μL of sterile deionized water. For *eae* confirmation, simplex PCR assays was carried out in a 25 μL volume containing, 5 μL of nucleic acid template, 12.5 μL of commercial Supreme PCR Master Mix 2x (NZYtech), 0.5 μL (0.4 pmol. μL$^{-1}$) each primer and 6.5 μL of sterile deionized water.

## Serotyping

The determination of O and H antigens was carried out using the method previously described by Guinée et al. (1981) [24] with all available O (O1 to O181) and H (H1 to H56) antisera produced in the Laboratorio de Referencia de *E. coli*—University of Santiago de Compostela (LREC-USC). To remove nonspecific agglutinins, all antisera were absorbed with cross-reacting antigens. Isolates that did not react with O antisera were classified as nontypeable (ONT) and those non motile were denoted as HNM.

Richness (S), Shannon's diversity (H), Simpson's diversity (D), and Simpson's evenness (E) [25] were calculated using the serotype data obtained for both heifers and lactating cows. Serotypes diversity of the STEC isolates obtained from lactating cows and heifers was determined using Simpson's numerical index (D) as described by Hunter and Gaston [25]. Values for *D* range between 0 and 1, with a value of 1 indicating the most diverse population. *D* depends on the number of serotypes identified by the test method and the relative frequencies of these serotypes and was calculated by the following formula:

$$D = 1 - \frac{\sum_{j=1}^{s} n_j(n_j - 1)}{N(N-1)}$$

where N is the total number of unrelated isolates in the sample population, *s* is the total number of types described, and *nj* is the number of strains that belong to the *j* the type.

## Results and discussion

### STEC prevalence in cattle

A total of 409 animals were sampled in 21 dairy farms from the North of Portugal. A total of 133 STEC isolates were recovered from 112 positive animals, which, overall, gave a prevalence

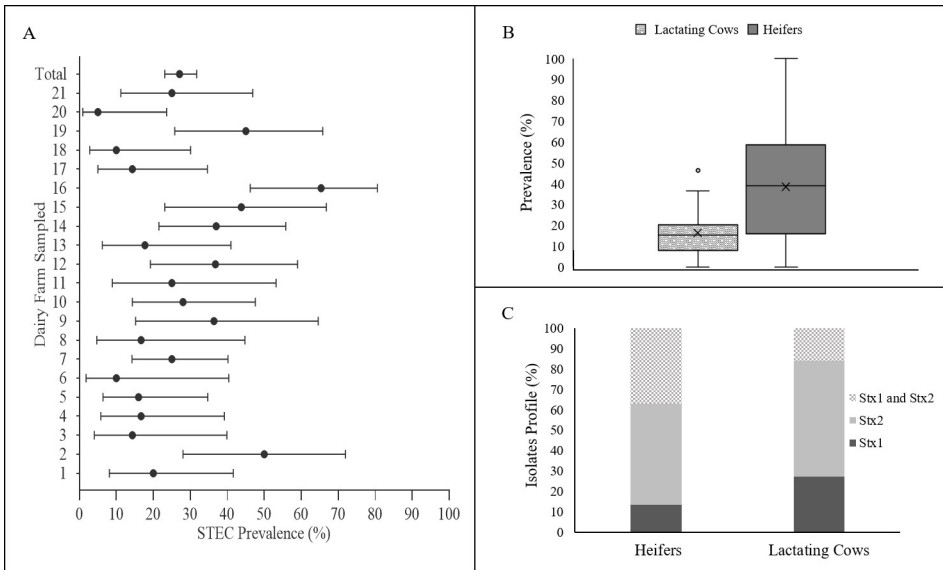

**Fig 2. Prevalence of Shiga toxin-producing *E. coli* (STEC) in the North of Portugal.** (A)- Prevalence of STEC per farm (error bars show 95% confidence intervals); (B) prevalence in heifers and lactating cows (box plots graph showing median, interquartile range and discrepant points); (C) isolates Stx profile per animal type.

rate of 27.4%. This observation is not surprising when comparing to similar prevalence studies. In Spain, between 1993 and 1995, the overall prevalence rates of STEC colonization were estimated in 37% for calves and 27% for cows [12]. These values were not far from the prevalence values found in Germany (18%) and France (34%) [26, 27]. In the US, a study conducted in Michigan in 2012 has shown STEC prevalence rates of 21% for beef cattle and 13% for dairy cattle [28]. In this work, the prevalence rates were calculated only considering the animals that resulted in the recovery of STEC isolates. Positive samples in enrichment (*stx* screening), but without STEC isolates, were disregarded. So, their prevalence is likely higher, as STEC is sometimes difficult to be isolated, especially if present in low concentrations. Another important observation is the number of STEC positive farms. All farms sampled in this study were positive for STEC presence, with prevalence rates per farm ranging from 5% to 65.4% (Fig 2A). The results are similar to that found by Blanco *et al.* (1996) [29] in Spain (STEC found on 84% of the farms and the proportion of animals infected varied from 0–63% and Venegas-Vargas *et al.* (2016) [28] in the United States, that have reached an overall prevalence of 13%, but with great variability between the sampled farms (6% to 54%). Many factors can be involved in the variability of prevalence rates between farms such as sensitivity and specificity of the methods used to detect STEC isolates, geographical area, year and annual sampling season, water supply, herd size, slurry and manure spreading on pasture, as well as the age of animals [30]. Animal age is usually a crucial factor. STEC were more commonly isolated from young animals, as demonstrated in the present study, where about 45% (70 out of 155) of the heifers were colonized by STEC in opposition to, only 16% of the adult cows (42 out of 254) ($p<0.05$, Fisher exact test statistic value is $<0.00001$) (Fig 2B) and this might explain the variations found between farms. This observation has already been reported in other studies [12, 31, 32]. A higher prevalence of STEC in young cattle has been related with a lower diversity of the gut microflora at early ages, which increases as the cattle matured [33]. Also, super-shedders of STEC have been found to be more common among younger animals [34]. Other important factors that can affect STEC prevalence are related with seasonality [35] or even with the immediate environment of the animal [18]. While the management systems were very similar

among farms, seasonality has not been taken into account in this study, due to the large sampling period needed to reach the expected sample size. So, these prevalence values can only be seen as an average prevalence data for this particular period. Regarding the *stx* genes found in STEC isolates 24 (18%) harboured only *stx1* genes, 69 (51.9%) possessed *stx2* genes, and 40 (30.1%) encoded both *stx1* and *stx2*. The Fig 2C shows the distribution of isolates obtained by heifers and lactating cows according to the presence of *stx* genes. Overall, differences were found in the number of isolates with only *stx1* genes and isolates carrying both *stx1* and *stx2* genes. For *stx1* gene, and while the same number of *stx1*-positive isolates have been found in heifers (12) and adult cows (12), *stx1* isolates have represented 27.3% (12/44) of the STEC isolates recovered from adult cows and 13.5% of the isolates recovered from heifers. In opposition, the number of *stx1*/*stx2*-positive strains were higher in heifers, a total of 33 out of 89 (37.1%), than in adult cows, with a total of 7 isolates out of 44 (15.9%). This might be close related with the diversity/serotypes of isolates that are more prevalent in the two groups of animals, as, for instance, more virulence serotypes have been consistently associated with *stx2* genes [36].

In association with Shiga toxins, the presence of the intimin gene (*eae*) is another relevant virulence gene commonly assessed in STEC. The intimin causes lesion in the intestine, in association with Shiga toxins and usually defines strains as enterohemorrhagic *E. coli* (EHEC) [37]. Thus, EHEC strains are associated with severe disease outcomes, including haemorrhagic colitis and HUS. Intimin (*eae*) virulence gene was detected in 29 STEC isolates (21.8%). The frequency of *eae* was similar in dairy cows and heifers. The fact that *eae*-positive STEC isolates were quite common is of great relevance as their pathogenic potential is even higher. While the presence of *eae* is not mandatory in isolates causing haemorrhagic colitis and HUS, its role in *E. coli* intimate attachment is very well established as an increased risk factor for human health [37, 38]. Assessment of *eae* presence in other cattle-STEC prevalence studies have reached to similar results, with 12% of the strains showing the presence of this virulence gene in healthy dairy cattle [36, 39]. In Spain the intimin (*eae*) virulence gene was detected in 29% of the bovine STEC isolates [12].

## Serotypes of STEC isolates

The serotype is another relevant property when assessing the virulence potential of STEC, as some specific ones are highly associated with food outbreaks. This is not only related with *stx* and *eae* presence, but most probably with the genetic pool of other virulence genes associated with those particular serotypes [40]. On this regard, it is important to bear in mind that the *E. coli* genome presents high plasticity. While *E. coli* strains usually contain about 5000 genes, only approximately 3000 compose the species core genome (since they are found in all *E. coli* genomes). The other ones represent the "accessory genome" that makes the huge heterogeneity and genetic diversity among *E. coli* strains [41, 42]. Because of this, serotype remains as a valuable tool for epidemiologic studies, not only for establishing the potential source/relationship of relevant isolates, but also to evaluate the clonal and pathogenic potential of isolates. Among the STEC, O157:H7 is the most well-known serotype due to its association with foodborne outbreaks; but there is a worldwide increase in cases of human disease related to non-O157. Serotyping of our STEC isolates identified 72 different O:H serotypes belonging to 40 O serogroups and 23 H (flagellar) types including non-motile (HNM) and non-typeable (NT) isolates (Table 2).

The serotypes Richness found in heifers was higher (50 serotypes) than that found in the milking cows (35 serotypes); nonetheless, Simpson indexes of diversity were very high in both populations (Table 3), either using only the serotypes data or the serotypes associated with the

**Table 2. Diversity and frequency of STEC serotypes isolated from dairy cattle (lactating cows and heifers) faeces.**

| Serotype | Number of isolates | Number of farms (farms ID) | Serotype | Number of isolates | Number of farms (farms ID) |
|---|---|---|---|---|---|
| O1:H6 | 1 | 1 (16) | O113:H36 | 1 | 1 (12) |
| **O1:H20** | 1 | 1 (12) | **O113:HNM** | 1 | 1 (18) |
| *O1:HNM* | 1 | 1 (7) | O115:H2 | 1 | 1 (17) |
| **O2:H27** | 1 | 1 (14) | O115:HNM | 1 | 1 (17) |
| **O6:H34** | 3 | 1 (16) | **O116:H21** | 4 | 3 (4; 7; 13) |
| O7, O103:H7 | 1 | 1 (4) | **O116:HNM** | 3 | 3 (6; 9; 10) |
| O8:H8 | 1 | 1 (15) | O116:HNT | 2 | 1 (2) |
| *O8:H14* | 1 | 1 (16) | **O119:H25** | 3 | 3 (4; 9; 18) |
| **O8:H19** | 1 | 1 (3) | **O119:HNM** | 1 | 1 (7) |
| *O15:H2* | 1 | 1 (12) | **O126:H20** | 1 | 1 (8) |
| O15:H16 | 1 | 1 (20) | O130:H47 | 1 | 1 (14) |
| *O15:HNM* | 1 | 1 (12) | O136:H12 | 4 | 2 (1; 5) |
| *O18:HNM* | 1 | 1 (15) | O136:HNM | 1 | 1 (5) |
| O19:H27 | 3 | 1 (14) | O140:HNT | 1 | 1 (10) |
| *O22:H8* | 3 | 2 (14;15) | O142:H34 | 1 | 1 (14) |
| *O26:H11* | 1 | 1 (7) | **O150:H8** | 1 | 1 (11) |
| O29:H1 | 1 | 1 (13) | O150:H21 | 1 | 1 (7) |
| O29:H12 | 20 | 5 (1; 2; 7; 16; 19) | *O157:H7* | 1 | 1 (10) |
| O32:H19 | 1 | 1 (2) | *O165:H25* | 3 | 2 (12; 14) |
| O39:H25 | 1 | 1 (15) | O166:HNM | 1 | 1 (7) |
| O55:H8 | 2 | 1 (10) | *O172:HNM* | 1 | 1 (11) |
| O55:H12 | 2 | 1 (16) | *O174:H21* | 1 | 1 (10) |
| O55:H21 | 1 | 1 (7) | O177:H34 | 1 | 1 (16) |
| *O55:HNM* | 1 | 1 (10) | O177:H44 | 1 | 1 (11) |
| O76:H21 | 1 | 1 (12) | **O177:HNM** | 4 | 4 (10; 13; 16; 17) |
| O88:H38 | 1 | 1 (2) | **O179:H8** | 2 | 1 (19) |
| O89:H38 | 1 | 1 (10) | O181:H19 | 1 | 1 (3) |
| O91:H8 | 4 | 2 (15; 16) | O183:H18 | 1 | 1 (8) |
| O91:H12 | 1 | 1 (2) | O183:HNM | 1 | 1 (9) |
| **O103:H6** | 1 | 1 (16) | **ONT:H8** | 1 | 1 (14) |
| **O103:H25** | 1 | 1 (16) | ONT:H11 | 2 | 1 (9) |
| O103:H34 | 1 | 1 (16) | ONT:H16 | 4 | 3 (4; 16; 17) |
| O109:H11 | 1 | 1 (15) | **ONT:H21** | 2 | 1 (4) |
| O109:H25 | 2 | 1 (16) | ONT:H25 | 1 | 1 (14) |
| O109:H34 | 1 | 1 (16) | ONT:H35 | 1 | 1 (3) |
| *O113:H21* | 8 | 4 (1; 7; 19; 21) | **ONT:HNM** | 2 | 2 (9; 14) |

The serotypes previously associated with human STEC are highlighted at bold and serotypes previously associated with human STEC causing HUS are highlighted at bold and a italic characters. NM = non-motile; NT = non-typeable.

**Table 3. Richness and diversity estimation for the STEC serotypes found in heifers and lactating cows.**

| | | | All STEC | STEC from lactating cows | STEC from Heifers (n = 89) |
|---|---|---|---|---|---|
| | | | (n = 133) | (n = 44) | |
| Richness | | | 72 | 35 | 50 |
| Simpson's index of diversity [*] | | Serotype | 0,969 | 0,989 | 0,948 |
| | | Serotype/Stx | 0,973 | 0,989 | 0,957 |

[*]Diversity index was calculated taking into account the serotypes data alone and the serotype combined with *stx*-profile.

*stx*-profile of the strains. This is probably due to the fact that serotype O29:H12 dominates in this population (representing 19 of the 50 serotypes found).

In fact, the most frequent serotypes was O29:H12 (15%), followed by O113:H21 (5.2%). The serotype O29:H12 was mainly found in heifer faeces (20 isolates) and was present at five different farms, while the seven isolates of serotype O113:H21 was distributed in four farms and it was found in heifer and lactating cow faeces (Table 2). The serotype O113:H21 is considered one of the relevant non-O157 STEC serotypes associated with severe human infections [26] and it has been reportedly associated with dairy cattle [43], but, to, to the best of our knowledge, serotype O29:H12 has not been previously associated with cattle faeces. Interestingly, 31 of the 72 serotypes found in the present study in bovine STEC isolates have been associated in previous studies with STEC isolates causing human infections and 13 of them were associated with haemolytic uremic syndrome [12, 38, 44–46]. Despite the clinical relevance of this data, the fact is that most of the studies evaluating the STEC in cattle has reached to similar observations, with studies reporting high numbers of different serotypes, ranging from 17 to 113 [12, 47], several of them associated with human infection.

## Conclusions

The main objective of the present study was to estimate the prevalence of STEC in dairy cattle in Portugal, and to evaluate their serotypes diversity. This could provide us with valuable information on the natural reservoir diversity, as well as their pathogenic potential, risks for consequent food contamination and human infection. Data has shown a high prevalence of STEC in dairy cattle (27%), in line with similar studies performed in European and non-European countries. All farms were positive for the presence of STEC and, as expected, heifers were most commonly colonized than adult cows. Serotypes O29:H12 and O113:H21, were found in at least four different farms. Many serotypes found in this study have been previously associated with serious diseases in humans, so the search for innovative tools for controlling these pathogens in animal husbandry is essential to prevent the spread to food and environment. Also, this prevalence information should be taken into account when designing the national surveillance programs for STEC.

## Supporting information

**S1 Table. Farms enrolled in the present study, their location, total number of animals and main feed.** [a]Total number of animals is the sum of lactating cows, dry cows and heifers. [b]Information provided by the farm owner.
(TIF)

**S2 Table. Conventional PCR primer and conditions for amplification of STEC virulence genes.**
(TIFF)

## Acknowledgments

We would like to thank topographer Fabiano Balem for making the map.

## Author Contributions

**Conceptualization:** Carlos Pinto, Hugo Oliveira, Gonçalo Almeida, Carina Almeida.

**Data curation:** Andressa Ballem, Soraia Gonçalves, Isidro Garcia-Meniño, Saskia C. Flament-Simon, Jorge Blanco, Gonçalo Almeida.

**Formal analysis:** Andressa Ballem, Isidro Garcia-Meniño, Jesús E. Blanco, Conceição Fernandes, Maria José Saavedra, Carlos Pinto, Hugo Oliveira, Jorge Blanco.

**Funding acquisition:** Hugo Oliveira, Carina Almeida.

**Investigation:** Andressa Ballem, Soraia Gonçalves, Isidro Garcia-Meniño, Saskia C. Flament-Simon, Jesús E. Blanco, Carlos Pinto, Jorge Blanco.

**Methodology:** Carlos Pinto, Jorge Blanco, Gonçalo Almeida.

**Project administration:** Carina Almeida.

**Supervision:** Conceição Fernandes, Maria José Saavedra, Jorge Blanco, Gonçalo Almeida, Carina Almeida.

**Writing – original draft:** Andressa Ballem.

**Writing – review & editing:** Conceição Fernandes, Maria José Saavedra, Hugo Oliveira, Jorge Blanco, Gonçalo Almeida, Carina Almeida.

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
