## [Decision Letter · Decision Letter 0]

11 Sep 2020

PONE-D-20-19097

Prevalence and Serotypes of Shiga Toxin-Producing Escherichia coli (STEC) in Dairy Cattle from Northern Portugal

PLOS ONE

Dear Dr. Oliveira,

Thank you for submitting your manuscript to PLOS ONE. After careful consideration, we feel that it has merit but does not fully meet PLOS ONE’s publication criteria as it currently stands. Therefore, we invite you to submit a revised version of the manuscript that addresses the points raised during the review process.

Please could you respond to all points raised by the two reviewers.

We look forward to receiving your revised manuscript.

Kind regards,

Dov Joseph Stekel

Academic Editor

PLOS ONE

Journal Requirements:

2.We note that [Figure(s) 1] in your submission contain [map/satellite] images which may be copyrighted. All PLOS content is published under the Creative Commons Attribution License (CC BY 4.0), which means that the manuscript, images, and Supporting Information files will be freely available online, and any third party is permitted to access, download, copy, distribute, and use these materials in any way, even commercially, with proper attribution. For these reasons, we cannot publish previously copyrighted maps or satellite images created using proprietary data, such as Google software (Google Maps, Street View, and Earth). For more information, see our copyright guidelines: http://journals.plos.org/plosone/s/licenses-and-copyright.

1.    You may seek permission from the original copyright holder of Figure(s) [1] to publish the content specifically under the CC BY 4.0 license. 

3.Thank you for stating the following in the Acknowledgments Section of your manuscript:

[IGM acknowledge the Consellería de Cultura, Educación e Ordenación

Universitaria, Xunta de Galicia for the grant ED481A-2015/149 and SCFS acknowledges

the FPU programme for the grant FPU15/02644 from the Secretaría General de

Universidades, Spanish Ministerio de Educación, Cultura y Deporte.]

 [Funding statment is presented in the manuscript body]

Reviewers' comments:

Reviewer's Responses to Questions

**Comments to the Author**

1. Is the manuscript technically sound, and do the data support the conclusions?

Reviewer #1: Partly

Reviewer #2: Partly

2. Has the statistical analysis been performed appropriately and rigorously? 

Reviewer #1: Yes

Reviewer #2: I Don't Know

3. Have the authors made all data underlying the findings in their manuscript fully available?

Reviewer #1: Yes

Reviewer #2: Yes

4. Is the manuscript presented in an intelligible fashion and written in standard English?

Reviewer #1: Yes

Reviewer #2: Yes

5. Review Comments to the Author

Reviewer #1: 1. Some concerns of the validity of the PCR results as the DNA extraction method has not been validated for use on faeces/soil which is more complex than the food substances that the method was designed for. However, if they have considered inhibition and included controls of spiked faecal material then those concerns are mitigated. Within the manuscript there is no statement regarding the inclusion of positive and negative controls during the DNA extraction process, so explanation of what they did will add clarity and confirm that the results are valid and sensitivity was not compromised by the chosen extraction method. More explanation could be given and the results and discussion section reads more as a statement of results in some places without proposed explanations for the findings. They also need to check whether they wanted 29 or 31 serotypes to have been placed in bold text in Table 2 as the table conflicts with the in text reference.

4. While the paper is largely well written, I have suggested some minor edits below to make it clearer to the reader.

Summary of paper

The paper examined the prevalence of Shiga Toxin-Producing Escherichia coli (STEC) and serotypes of STEC in heifers and dairy cattle in Northern Portugal. This research appeared to be novel due to being undertaken in North Portugal and the results compared with other similar studies in different locations. The authors identified that STEC was significantly more likely to be isolated from heifers than lactating cattle, as had been found in other studies (e.g. [1] which was not referenced). In total, 133 STEC were isolated from 409 cattle across 21 farms; 13.8% (n = 24) carried stx1, 39.7% (n = 69) carried stx2 and 23% (n = 40) carried both stx1 and stx2. The overall prevalence rate of STEC (27.4%) was similar to that identified in Spain, Germany and France in other studies that were referenced. The study identified 74 different O:H serotypes belonging to 40 O serogroups and 21 H groups including non-motile (HNM) and non-typable (NW) isolates; 29 of the identified serotypes were stated (in text) to have been linked to foodborne outbreaks.

[1] 10. Mir RA, Weppelmann TA, Kang M, Bliss TM, DiLorenzo N, Lamb GC, et al. Association between animal age and the prevalence of Shiga toxin-producing Escherichia coli in a cohort of beef cattle. Vet Microbiol. 2014

Introduction

• Rephrase line 68 – ‘ STEC are found in a wide variety of animal species as a natural gut inhabitant.’

• Lines 69-72: Rephrase to demonstrate that the food is likely contaminated with faeces containing STEC, rather than saliva/blood etc. (e.g. ref: Fairbrother & Nadeau, 2006).

• Line 78-9: ‘investigate the prevalence of STEC in healthy dairy cattle…’ . Do you need to add ’and heifers’ as they were referred to separately in the abstract (L29).

Methods

Methods

• Line 120 – do not need ‘the’ in that sentence.

• Line 123 – there isn’t a ‘quantitative’ aspect to the PCR then I would refer to it as presence-absence real time PCR (e.g. PA-PCR)

• Line 123 – sensitivity of the method would be good to include and any discussion surrounding the potential for inhibition by other components from the enriched faeces or reference the method as being suitable to avoid inhibition by compounds (such as humic acids) that may decrease sensitivity.

• Line 134 – No controls discussed for the extraction of DNA from enriched faecal slurry. Should include information regarding a positive and negative control for the extraction process as well as the PCR plate set-up, particularly due to the use of an extraction method not designed for faeces.

Results and discussion

• Line 185 – Remove ‘in fact’.

• Line 189 ‘in the Michigan’, remove ‘the’.

• Line 191-192 – requires different phrasing.

• The section describing stx1 genes as being more common in adults than heifers, while those STEC carrying both genes were more frequently found in heifers requires more discussion. In conjunction with this I think that the section stating that STEC was more commonly isolated from heifers (45% heifers, 16% of adults) – already reported in other studies. So, despite STEC being more frequently isolated from heifers, stx1 was still more common in the adult population? This section reads more like a statement of results and could do with more discussion and tying in with other literature (Lines 203-214). Are there any farm metadata or other studies that could provide insight?

• Line 228- ‘well established’.

• Line 228 – missing references.

• Line 243 – ‘establishing’

• Line 245 – ‘foodborne’

• Higher diversity of serotypes from heifers (55) compared to lactating (34 serotypes) but no discussion.

• In text it is stated that 29/74 (line 260) but 31 are indicated in bold (Table 2).

• More could be done to achieve the objective of evaluating the serotype diversity (line 266). Did some farms/areas have higher diversity than others? Do certain serotypes appear to out compete others? Is it possible to run Simpson/Shannon diversity on these data? Did the farms with high or low diversity have high cattle numbers or an older cattle cohort?

Conclusion

• Did not mention that Portugal does not have a surveillance system in place for monitoring STEC/VTEC infections like most EU/EEA Member States – could be worth mentioning in the conclusion to make a case for increased surveillance?

Reviewer #2: The reviewer feel that the paper contains valuable scientific research which would benefit people in similar or closely related field. However, the paper would require a bit of some work prior to publication, areas of which are listed below followed by specific areas:

1. There is need for though proof reading to correct on grammar and some tenses.

2. Detailed explanation why there were variations on STEC prevalence in Heifers and cows

3. How did the study consider factors such as season, management systems between farms etc during sample collection? Any adjustments made for these factors among others so as to avoid bias.

Abstract

Include what this - stx1 and stx2 stands for at the start then can be followed by use of short form.

The prevalence of STEC in heifers was significantly higher than in lactating cows (include p value)

Introduction

51-52 – Did you mean third place in Europe?

78 - Revisit the wording on how you have structured the aim at the start of the sentence.

Materials and methods

83-91 – No need to include the names of the persons

93: check on tenses and also review the sentence

98: what criteria was used for deciding to work with farms >50 herd size?

100-101: though mentioned in the abstract the clarity on type of samples collected not shown

105: software for mapping should not be included on the figures section instead should be somewhere on the materials and methods.

111: neighbour?

Results and discussion

Information why the STEC prevalence was higher in heifers than milking cow is lacking in this work.

Also, the study needs to show how it took care of animals kept outside compared to those managed indoors. No clear information whether the animals were indoors throughout or were at times given the opportunity to graze outside during warmer months etc?

Worth detailing why the stx1 isolates was more common in adult cows compared to heifers;

214 however, the proportion of isolates harbouring both genes were higher in heifers.

How did the study consider factors such as season? This needs to be detailed in the stud

6. PLOS authors have the option to publish the peer review history of their article (what does this mean?). If published, this will include your full peer review and any attached files.

Reviewer #1: **Yes: **Sian Mari Powell

Reviewer #2: No

---

## [Author Response · Author response to Decision Letter 0]

25 Nov 2020

We have incorporated the suggestions made by the editor and reviewers. Those changes are highlighted within the Marked Up manuscript as well in the Response to Reviewers letter that we attached.

---

## [Editor Report · Decision Letter 1]

16 Dec 2020

Prevalence and Serotypes of Shiga Toxin-Producing Escherichia coli (STEC) in Dairy Cattle from Northern Portugal

PONE-D-20-19097R1

Dear Dr. Oliveira,

We’re pleased to inform you that your manuscript has been judged scientifically suitable for publication and will be formally accepted for publication once it meets all outstanding technical requirements.

Kind regards,

Dov Joseph Stekel

Academic Editor

PLOS ONE
---

## [Editor Report · Acceptance letter]

21 Dec 2020

PONE-D-20-19097R1 

Prevalence and Serotypes of Shiga Toxin-Producing *Escherichia coli* (STEC) in Dairy Cattle from Northern Portugal 

Dear Dr. Oliveira:

I'm pleased to inform you that your manuscript has been deemed suitable for publication in PLOS ONE. Congratulations! Your manuscript is now with our production department. 

Kind regards, 

on behalf of

Dr. Dov Joseph Stekel 

Academic Editor

PLOS ONE